# Assessing Passengers’ Motion Sickness Levels Based on Cerebral Blood Oxygen Signals and Simulation of Actual Ride Sensation

**DOI:** 10.3390/diagnostics13081403

**Published:** 2023-04-12

**Authors:** Bin Ren, Qinyu Zhou

**Affiliations:** Shanghai Key Laboratory of Intelligent Manufacturing and Robotics, School of Mechatronic Engineering and Automation, Shanghai University, Shanghai 200444, China; zqy1101@shu.edu.cn

**Keywords:** motion sickness, functional near-infrared spectroscopy (fNIRS), cerebral oxygenation signals, Bayesian ridge regression (BRR), Graybiel scale, vehicle test

## Abstract

(1) Background: After motion sickness occurs in the ride process, this can easily cause passengers to have a poor mental state, cold sweats, nausea, and even vomiting symptoms. This study proposes to establish an association model between motion sickness level (MSL) and cerebral blood oxygen signals during a ride. (2) Methods: A riding simulation platform and the functional near-infrared spectroscopy (fNIRS) technology are utilized to monitor the cerebral blood oxygen signals of subjects in a riding simulation experiment. The subjects’ scores on the Fast Motion sickness Scale (FMS) are determined every minute during the experiment as the dependent variable to manifest the change in MSL. The Bayesian ridge regression (BRR) algorithm is applied to construct an assessment model of MSL during riding. The score of the Graybiel scale is adopted to preliminarily verify the effectiveness of the MSL evaluation model. Finally, a real vehicle test is developed, and two driving modes are selected in random road conditions to carry out a control test. (3) Results: The predicted MSL in the comfortable mode is significantly less than the MSL value in the normal mode, which is in line with expectations. (4) Conclusions: Changes in cerebral blood oxygen signals have a huge correlation with MSL. The MSL evaluation model proposed in this study has a guiding significance for the early warning and prevention of motion sickness.

## 1. Introduction

As the automobile industry continues to grow, the problem of motion sickness among passengers has also emerged. As an important evaluation index, ride comfort has attracted more and more attention from consumers. After motion sickness occurs in the process of riding, this can easily cause passengers to have a poor mental state, cold sweats, nausea, and even vomiting symptoms. Although the symptoms of motion sickness are usually short-lived and dissipate sometime after stopping the ride, they can affect the passenger’s normal life. In addition, for those in poor health, motion sickness may induce underlying chronic diseases, resulting in secondary injuries. It is very difficult to solve the problem of motion sickness from the source because different individuals have different causes of motion sickness. Noise, light intensity, temperature, ventilation, passengers’ own psychological state, and drivers’ driving style are all possible reasons for motion sickness [1]. However, the human symptoms caused by motion sickness are the same. These symptoms mainly manifest as cold sweats, dizziness, nausea, vomiting, etc. Therefore, the current research on motion sickness mainly reflects passengers’ motion sickness level (MSL) by monitoring various physiological indicators. By monitoring physiological changes in real time, the occurrence of motion sickness symptoms can be predicted and intervention measures can be introduced to effectively avoid or alleviate the appearance of motion sickness [2]. In this study, a ride simulation platform is built to restore the actual ride sensation. The measurement method of cerebral blood oxygen signals from motion sickness is established using near-infrared equipment to quantitatively evaluate and predict MSL.

## 2. Background

### 2.1. Definition and Causes of Motion Sickness

Irwin first proposed the concept of motion sickness in 1881 [3]. With the deepening of relevant scholars’ research, motion sickness is currently defined as a general term for the human body’s false perception of motion state and a series of physiological reactions caused by various factors. This concept is equivalent to the phenomenon of carsickness and seasickness in daily life. The causes of motion sickness come from different factors, and there are differences among individuals. The existing theories to explain motion sickness include the central nervous system dysfunction theory, the endocrine dysfunction hypothesis, the sensory conflict theory, etc. The central nervous system dysfunction theory defines motion sickness as a stress response of the central nervous system due to neurotransmitters [4,5]. The hypothesis of abnormal endocrine function refers to the belief that when people are overstimulated, the endocrine hormones in plasma are abnormal, resulting in the reduction in gastrointestinal motility, and vomiting and nausea [6]. Sensory conflict theory is currently the most accepted explanation for the source of motion sickness [7]. According to sensory conflict theory, motion sickness is caused by the mismatch of otolith information that senses gravity and linear acceleration, semicircular canal information that senses angular acceleration, visual information, and proprioceptive information [8]. For example, passengers browse static content on their mobile phones in the car, but their bodies perceive the dynamic movement of the car as it drives. That is why most people think it is easy to become dizzy when using a phone in a car.

### 2.2. Recognition and Classification of Motion Sickness

When passengers suffer from motion sickness, their physiological states and behavioral performance change accordingly. At present, the detection method of motion sickness is mainly through subjective scale evaluation and physiological signal analysis. The Simulator Sickness Questionnaire (SSQ) can be utilized to evaluate MSL [9]. The Motion Sickness Susceptibility Questionnaire (MSSQ) is regarded as a feasible method to distinguish the motion sickness susceptibility of the subjects [10]. According to different MSSQ values, subjects can be divided into multiple groups for controlled experiments [11]. Relevant scholars designed the Visually Induced Motion Sickness Susceptibility Questionnaire (VIMSSQ), which is specifically used for the assessment of visually induced motion sickness [12]. The Fast Motion sickness Scale (FMS) can be completed in real time during the process of the experiment. FMS can obtain the subjective motion sickness evaluation value of the subjects in the process of the experiment without affecting the experiment [13,14]. In addition, the Vertigo Symptom Scale (VSS) and Misery Scale (MISC) have been proven to be effective for the representation of MSL in studies [15,16].

In the evaluation of physiological signals, the method of judging MSL by EEG signals is the most accepted. The absolute power of EEG signals from different frequency bands is closely related to MSL. The absolute power of the delta band increases significantly and that of the alpha band decreases significantly in patients with motion sickness [17]. In the study of Chuang et al., it was pointed out that the higher the degree of motion sickness, the higher the activation degree of alpha and gamma frequency bands, including the motor area, parietal area, and occipital area [18]. The pupil rhythm detection method based on an infrared camera can also reflect the occurrence of motion sickness. After the subjects experience motion sickness, the mean and standard deviation of pupil diameter increases, and the coherence ratio of pupil rhythm decreases [19]. In addition, physiological signals such as blood pressure and respiration are closely related to motion sickness [20]. For motion sickness induced by visual stimuli, Li Y et al. extracted subjects’ EEG, center of pressure, and head and waist motion trajectories as input features. The K-Nearest Neighbor classifier, logistic regression, random forest, and multilayer perceptron neural network were considered as classifiers. The results showed that this algorithm was effective for both binary and ternary classification of motion sickness [21]. Recently, M et al. quantified the interaction between muscle activation, brain activity, and cardiac behavior during motion sickness in their study [22]. Ko L et al. proposed an extended version of the inheritable bi-objective combinatorial genetic algorithm. Compared with general prediction models, this algorithm improves the test accuracy by 10% to 20% for reporting the level of motion sickness [23]. The research of these scholars shows that the characteristics of physiological signals combined with machine learning and deep learning can effectively identify the degree of motion sickness.

### 2.3. Motion Sickness Recognition Based on Cerebral Blood Oxygen Signals

Since the pathogenesis of motion sickness is often explained by sensory conflict theory, physiological changes in the brain are important for the analysis of motion sickness. Brain imaging technology has been widely accepted to record brain activity associated with motion sickness symptoms [24]. Functional near-infrared spectroscopy (fNIRS), with better spatial resolution, is a reliable technique to monitor brain activity during motion sickness. Based on the six-degree-of-freedom driving simulator and the fNIRS device, Zhang et al. collected the driving operation data and cerebral blood oxygen data of the participants. The results showed that the occurrence of motion sickness was related to the occipital lobe, indicating the correlation mechanism between motion sickness and cerebral cortex activity [25]. Kinoshita et al. created two stereoscopic video clips with different background elements and measured regional cerebral blood flow while viewing stereoscopic video clips by the fNIRS technique. The results showed that the differences in background elements in the peripheral visual field during stereoscopic video clips affected the regional cerebral blood flow from the occipital lobe to the prefrontal cortex [26]. Takada et al. used fNIRS techniques to measure cerebral blood flow when subjects viewed stereoscopic video clips with and without background. During the observation of the two background cases, the concentration of oxygenated hemoglobin in the occipital lobe increased significantly [27]. Hoppes et al. applied fNIRS technology to explore brain activation during optical flow. The results showed that greater brain activation was observed in the bilateral frontotemporal parietal lobes when the optical flow was finally observed. The optical flow activated the bilateral frontotemporal parietal regions of the cerebral cortex. This activation was greater when the optical flow and fixed targets were observed [28].

From the research mentioned above, most scholars currently study vision-induced motion sickness through fNIRS technology. However, in the actual ride process, few passengers will cause their own motion sickness reaction because of the moving view out of the window. Therefore, this study restores the actual ride sensation through the ride simulation platform. The relationship between motion sickness and cerebral blood oxygen signals is studied by simulating real riding conditions.

### 2.4. Motivation and Structure of This Study

The purpose of this study is to investigate the link between motion sickness and cerebral blood oxygen data by building a ride simulation platform to restore the actual road conditions. This study has an enlightenment effect on the early warning and prevention of motion sickness. The remainder of the paper is organized as follows. The third section describes the design and process of the experiment. The fourth part introduces the preprocessing of the light intensity signals, the algorithm of transforming light intensity signals into cerebral blood oxygen signals, and the parameters of cerebral blood oxygen signals used for the recognition of motion sickness. After these cerebral blood oxygen parameters are tested, combined with the Bayesian ridge regression (BRR) algorithm, a prediction model for MSL is constructed in the fifth part. The sixth part introduces the process of the vehicle test, which is used to verify the validity of the motion sickness prediction model. The seventh part analyzes the advantages and disadvantages of this study and discusses the relevant measures of motion sickness alleviation and other research directions worthy of further study. Section 8 is the conclusion, which summarizes the contributions of this study.

## 3. Design of the Ride Simulation Experiment of Passengers

The study of motion sickness needs to be carried out in a safe and controllable environment, such as experiments on riding simulators [29]. Therefore, this study used a riding simulator built in the laboratory based on a six-degree-of-freedom platform. Table 1 shows the key parameters of the six-degree-of-freedom platform. The fNIR 2000C Imager device was utilized to monitor the brain status of the subjects in real time during the experiment. fNIRS is a safe and non-invasive method for brain state monitoring. As shown in Figure 1, the fNIR 2000C Imager device has two emitters and six detectors to monitor changes in prefrontal oxygenated and deoxygenated hemoglobin in real time.

Eight subjects were recruited in this study. These eight subjects were undergraduate students from Shanghai University, including four male students and four female students. Informed consent was obtained from the eight subjects before conducting the experiment.

Before the experiment, the experimenter guided the subjects to ride on the simulation platform and put on the fNIRS equipment for the subjects, as shown in Figure 2. The subjects were asked to complete the MSSQ questionnaire before the experiment. According to the MSSQ scores, the subjects were divided into three groups: insensitive to motion sickness, general, and sensitive to motion sickness. After starting the experiment, the experimenter was required to record the FMS scores (0–10 points) of the subjects by verbal interrogation every 1 min. A score of 0 indicated no motion sickness at all, while a score of 10 indicated severe motion sickness. In this way, the eight subjects formed a total of 80 sets of data. In the experiment, the behavior of the subjects was not constrained. The subjects were free to consult their mobile phones and communicate during the experiment to align their behavior with that of the actual ride.

At the end of the experiment, subjects were required to complete the Graybiel scale. The Graybiel scale belongs to a multidimensional symptom scale. Graybiel et al. identified nausea, pale skin, cold sweat, increased saliva, and drowsiness as the five symptoms of motion sickness [30]. There are also two types of symptoms: headache and dizziness. According to the different degrees of each symptom, different points are assigned, and finally the total score is obtained. According to the total score, the degree of motion sickness is divided into five grades: mild discomfort, moderate discomfort (B), moderate discomfort (A), severe discomfort, and motion sickness.

During the test, if the subject requested to terminate the experiment because of discomfort, the experimenter immediately suspended the movement of the platform and ended the experiment.

## 4. Calculation and Preprocessing of Cerebral Blood Oxygen Signals

The principle of using fNIRS for cerebral blood oxygen monitoring is that biological tissues have the characteristics of high scattering and low absorption for near-infrared light. Light of near-infrared bands can penetrate biological tissues to a sufficient depth. The hemoglobin of the human body carries chromophores, which have a strong absorption effect on near-infrared light. The change in the concentration of oxygenated hemoglobin and deoxygenated hemoglobin in human tissue causes a change in the absorption spectrum of light in human tissue. Therefore, the changes in cerebral blood oxygen signals can be reflected by detecting the near-infrared light intensity signals [31,32].

To preprocess the collected raw near-infrared light intensity data, firstly, some data segments with obvious local interference are deleted through observation. A 20-order low-pass FIR filter of 2 Hz is utilized to filter out noisy data. In the process of signal acquisition, the subject’s head will inevitably move. Therefore, the filtered light intensity data still have the interference of motion artifacts. Motion artifacts during the measurement can cause coupling or pressure changes between the light source and the detector. This change in pressure shows up as a burst of noise in the original light intensity signals. This kind of noise is different from the light intensity signals associated with regular cortical activities. This study refers to the Sliding-window Motion Artifact Rejection (SMAR) algorithm of Hasan Ayaz et al., which removes motion artifacts from the original light intensity data. SMAR algorithm flow is as follows [33].

The original near-infrared intensity signals are defined as $x_{\lambda }\left(n\right)$ where λ represents the wavelength and x represents the measurement position. Calculate a local coefficient of variation (CV) for each n as follows.
(1)$${\mathrm{CV}}_{d}\left(n\right)=\frac{\sqrt{\frac{1}{N}\sum_{j=n-N/2}^{j=n+N/2}{{(x}_{d}\left(j\right)-\frac{1}{N+1}\sum_{i=n-N/2}^{i=n+N/2}x_{d}\left(i)\right)}^{2}}}{\frac{1}{N+1}\sum_{i=n-N/2}^{i=n+N/2}x_{d}\left(i\right)}$$
where N+1 represents the window sample size. According to the calculation of ${\mathrm{CV}}_{d}\left(n\right)$, ${\mathrm{CV}}_{\lambda 1}\left(n\right)$ and ${\mathrm{CV}}_{\lambda 2}\left(n\right)$ can be obtained. Then calculate cleaned signals ${\;\hat{x}}_{d}$, ${\hat{x}}_{\lambda 1}$, and ${\hat{x}}_{\lambda 2}$.
(2)$${\hat{x}}_{d}(n)=\left\{\begin{array}{l} x_{d}{(n),\;\mathrm{CV}}_{d}{(n)\;<\;\tau }_{d}^{\mathrm{upper}}\\ \mathrm{NaN},\;\mathrm{else}\; \end{array}\right.$$
where NaN means ‘Not a Number’, which represents the excluded value. $\tau_{d}^{\mathrm{upper}}$ indicates the upper threshold value for dark current.
(3)$$X_{\lambda }^{n}=\frac{1}{N+1}\sum_{i=n-N/2}^{i=n+N/2}x_{\lambda }\left(i\right)$$
where $X_{\lambda }^{n}$ indicates the window mean centered around n.
(4)$${\hat{x}}_{\lambda }\left(n\right)=\left\{\begin{array}{c} {\mathrm{NaN},\;X}_{\lambda }^{n}{\;>\;s\;\mathrm{and}\;\tau }_{\lambda }^{\mathrm{lower}}{\;>\;\mathrm{CV}}_{\lambda }(n)\\ {\mathrm{NaN},\;\mathrm{CV}}_{\lambda }{\;>\;\tau }_{\lambda }^{\mathrm{upper}}\\ x_{\lambda }(n),\;\mathrm{else} \end{array}\right.\;$$
where $\tau_{\lambda }^{\mathrm{upper}}$ represents the upper threshold. $\tau_{\lambda }^{\mathrm{lower}}$ indicates the lower threshold value for the respective wavelength. s means range of saturation.

Finally, the light intensity signals are converted into cerebral blood oxygen signals through the modified Beer–Lambert law. The Beer–Lambert law is regarded as a description of the light propagation in biological tissues [34,35]. However, this law ignores the scattering of light in biological tissues. Because of the strong scattering characteristics of biological tissues, the scattering attenuation of biological tissues to light is much greater than the absorption attenuation. During the propagation of photons in biological tissues, multiple scattering occurs, and the scattering process is random. Therefore, Cope and Delpy et al. [36] introduced the differential path factor (DPF) to correct the actual optical path of photons in the modified Beer–Lambert law.
(5)$$A=\log\left(\frac{I_{0}}{I}\right)=\mathrm{DPF}(\lambda )\cdot \varepsilon^{\lambda }\cdot r\cdot C+G$$
where A represents absorbance. $I_{0}$ is the intensity of the incident light after SMAR. I is the intensity of the outgoing light. $\varepsilon^{\lambda }$ represents the absorption coefficient of a substance when the wavelength is λ. r is the distance between the light source and the photodetector. C is the concentration of the tested substance. G represents the attenuation caused by the absorption of other substances.

When two kinds of light-absorbing substances, HbO and HbR, are present in tissues, the following conclusions can be drawn.
(6)$$A=\log\left(\frac{I_{0}}{I}\right){=(\varepsilon }_{\mathrm{HbO}}^{\lambda }\cdot C_{\mathrm{HbO}}{+\varepsilon }_{\mathrm{HbR}}^{\lambda }\cdot C_{\mathrm{HbR}})\cdot \mathrm{DPF}(\lambda )\cdot r+G$$

Set a reference state. The changes in HbO and HbR concentrations can then be detected.
(7)$$\Delta A=\log\left(\frac{I}{I^{\prime }}\right)=\left(\varepsilon_{\mathrm{HbO}}^{\lambda }\cdot \Delta C_{\mathrm{HbO}}{+\varepsilon }_{\mathrm{HbR}}^{\lambda }\cdot \Delta C_{\mathrm{HbR}}\right)\cdot \mathrm{DPF}\left(\lambda \right)\cdot r$$
where ΔA is the change in absorbance. $\Delta C_{\mathrm{HbO}}$ and $\Delta C_{\mathrm{HbR}}$ represent the changes in the concentration of HbO and HbR, respectively. I and $I^{\prime }$ denote the outgoing light intensity in the reference state and after the reference state, respectively. When two different wavelengths of light are chosen, the following formula can be obtained.
(8)$$\Delta A^{\lambda_{1}}=\log\left(\frac{I_{\lambda_{1}}}{I_{\lambda_{1}}^{\prime }}\right)=(\varepsilon_{\mathrm{HbO}}^{\lambda_{1}}\cdot \Delta C_{\mathrm{HbO}}{+\varepsilon }_{\mathrm{HbR}}^{\lambda_{1}}\cdot \Delta C_{\mathrm{HbR}})\cdot {\mathrm{DPF}(\lambda }_{1})\cdot r$$
(9)$$\Delta A^{\lambda_{2}}=\log\left(\frac{I_{\lambda_{2}}}{I_{\lambda_{2}}^{\prime }}\right){=(\varepsilon }_{\mathrm{HbO}}^{\lambda_{2}}\cdot \Delta C_{\mathrm{HbO}}{+\varepsilon }_{\mathrm{HbR}}^{\lambda_{2}}\cdot \Delta C_{\mathrm{HbR}})\cdot {\mathrm{DPF}(\lambda }_{2})\cdot r$$

The values of $\Delta C_{\mathrm{HbO}}$ and $\Delta C_{\mathrm{HbR}}$ can be obtained by combining Equations (8) and (9). Here, for the convenience of calculation, the DPF values at different wavelengths are approximately equal. The changes in the concentration of HbO and HbR can finally be obtained.
(10)$$\Delta C_{\mathrm{HbO}}=\frac{\log\left(\frac{I_{\lambda_{1}}}{I_{\lambda_{1}}^{\prime }}\right)\cdot \varepsilon_{\mathrm{HbR}}^{\lambda_{2}}-\log\left(\frac{I_{\lambda_{2}}}{I_{\lambda_{2}}^{\prime }}\right)\cdot \varepsilon_{\mathrm{HbR}}^{\lambda_{1}}}{\mathrm{DPF}\cdot r\cdot {(\varepsilon }_{\mathrm{HbR}}^{\lambda_{2}}\cdot \varepsilon_{\mathrm{HbO}}^{\lambda_{1}}-\varepsilon_{\mathrm{HbR}}^{\lambda_{1}}\cdot \varepsilon_{\mathrm{HbO}}^{\lambda_{2}})}$$
(11)$$\Delta C_{\mathrm{HbR}}=\frac{\log\left(\frac{I_{\lambda_{1}}}{I_{\lambda_{1}}^{\prime }}\right)\cdot \varepsilon_{\mathrm{HbO}}^{\lambda_{2}}-\log\left(\frac{I_{\lambda_{2}}}{I_{\lambda_{2}}^{\prime }}\right)\cdot \varepsilon_{\mathrm{HbO}}^{\lambda_{1}}}{\mathrm{DPF}\cdot r\cdot {(\varepsilon }_{\mathrm{HbR}}^{\lambda_{1}}\cdot \varepsilon_{\mathrm{HbO}}^{\lambda_{2}}-\varepsilon_{\mathrm{HbR}}^{\lambda_{2}}\cdot \varepsilon_{\mathrm{HbO}}^{\lambda_{1}})}$$

## 5. Establishment of Motion Sickness Evaluation Model for Passengers

### 5.1. Extraction of Multiple Characteristic Parameters from Cerebral Blood Oxygen Signals

In this study, six characteristic parameters were extracted from the cerebral blood oxygen signals of the subjects to characterize their MSL values. The six characteristic parameters are $\Delta C_{HbO}$, $C_{HbO}$, $C_{HbR}$, $C_{HbT}$, $C_{\Delta Oxy}$ and ΔCOE.

$\Delta C_{HbO}$ is obtained directly from Formula (10), which is calculated from the light intensity signals by the Beer–Lambert law.

$C_{HbO}$ and $C_{HbR}$ are obtained by adding $\Delta C_{HbO}$ and $\Delta C_{HbR}$ to the reference state value, respectively.

$C_{HbT}$ and $C_{\Delta Oxy}$ represent the sum and difference of $C_{HbO}$ and $C_{HbR}$, respectively.

Here, $C_{HbT}$ represents total hemoglobin, which is calculated as follows.
(12)$$C_{HbT}=C_{HbO}+C_{HbR}$$
where $C_{\Delta Oxy}$ represents the difference between $C_{HbO}$ and $C_{HbR}$, which is calculated as follows.
(13)$$C_{\Delta Oxy}=C_{HbO}-C_{HbR}$$

ΔCOE is a valid index of local brain activity, and the following equation can be used to calculate ΔCOE.
(14)$$\Delta COE=\frac{\Delta C_{HbR}-\Delta C_{HbO}}{\sqrt{2}}$$

ΔCOE is an indicator of changes in vascular oxygenation, which can reflect neural activity. A rising ΔCOE value indicates intravascular hypoxia. A decreasing ΔCOE value indicates a high level of oxygenation in the blood vessels. Based on the motion sickness experiment on the driving simulator, Zhang et al. considered ΔCOE as the research object to explore the responses of different brain regions of the subjects in the straight driving condition. This study also proved the effectiveness of ΔCOE in evaluating the motion sickness state of drivers [26].

### 5.2. Normality Test and Correlation Test of the Parameters

Figure 3 reports the statistics of $\Delta C_{HbO}$ and $\Delta C_{HbR}$ of a subject in a 10 min riding simulation experiment. The data statistics graph shows the relationship between the ordered distribution and the standard normal distribution of all $\Delta C_{HbO}$ and $\Delta C_{HbR}$ values. As shown in Figure 3, the values of $\Delta C_{HbO}$ and $\Delta C_{HbR}$ satisfy the test of the normal Q-Q plot. Therefore, $\Delta C_{HbO}$ and $\Delta C_{HbR}$ are proven to be able to represent MSL, and the extreme values of $\Delta C_{HbO}$ and $\Delta C_{HbR}$ are random from the normal distribution.

In this study, the regression analysis method was utilized to predict MSL during the ride. The multiple characteristic parameters of cerebral blood oxygen signals are extracted, and these characteristic parameters need to be tested before regression prediction. The preconditions that need to be met to carry out the regression analysis are as follows. (1) The independent variable should have a linear correlation with the dependent variable, and the independent variable should be significant for the dependent variable. (2) It is required for there to be no multicollinearity among the independent variables. (3) The residual follows the normal distribution. (4) The residual error satisfies homogeneity of variance.

It can be seen from Figure 3 that the cerebral blood oxygen signals are subject to the normal distribution (the curves of the parameters are close to the theoretical normal distribution curves). Therefore, Pearson correlation coefficient was used to represent the correlation between the independent variables and the dependent variables as well as the correlation among the independent variables. Figure 4 reports the Pearson correlation coefficients of the six cerebral blood oxygen parameters and the subjective motion sickness evaluation scores. The significance test results are also displayed in Figure 4. In the significance test chart, the *p*-value not indicated means that the correlation is significant (*p* < 0.05). It can be seen from Figure 4 that the Pearson correlation coefficients of $C_{HbO}$, $C_{\Delta Oxy}$, and ΔCOE with the subjective evaluation score of motion sickness are 0.35, 0.38, and 0.24, respectively, showing significant correlation (*p* < 0.05). The Pearson correlation coefficients of $C_{HbR}$, $C_{HbT}$, and $\Delta C_{HbO}$ with the subjective evaluation score of motion sickness are 0.043, 0.21, and −0.19, respectively, and the correlation is not significant (*p* > 0.05). Therefore, $C_{HbO}$, $C_{\Delta Oxy}$, and ΔCOE are retained as the independent variables in the regression model construction.

However, the correlations among $C_{HbO}$, $C_{\Delta Oxy}$, and ΔCOE are significant (Figure 4). It indicates the existence of multicollinearity among the independent variables. The multicollinearity among independent variables can easily lead to the distortion of model estimation or make it difficult to estimate accurately. Thus, it is necessary to avoid multicollinearity in the regression process. This paper introduces a Bayesian ridge regression algorithm to address the multicollinearity issue.

### 5.3. Bayesian Ridge Regression Algorithm

Ridge regression is a regularization method often used in the regression analysis of ill-posed problems, which can solve the problem of the high correlation of independent variables in the regression process. Ridge regression solves the problem of multicollinearity by shrinking the parameter. It is a complement to least squares regression. It loses the unbiasedness in exchange for high numerical stability, thus obtaining high computational accuracy. Bayesian ridge regression (BRR) is a machine learning regression algorithm based on Bayesian theory. The functional formula of the Bayesian linear regression is as follows.
(15)$$y\left(x,\omega \right)=\overset{n}{\underset{j=0}{\sum }}\omega_{j}\psi_{j}\left(x\right)=\omega^{T}\psi \left(x\right)$$

The purpose of Bayesian regression is to find the distribution of parameter vectors with the minimum loss function. The loss function is given as follows.
(16)$$J\left(\omega \right)=\overset{m}{\underset{i=1}{\sum }}{\left\{y\left(x_{i},\omega \right)-t_{i}\right\}}^{2}$$
where n is the sample space dimension. m is the sample size. ω is the vector of parameters. $\psi \left(x\right)$ is a non-linear function of the input vector x.
(17)$$\psi_{0}\left(x\right)=1$$
(18)$$t_{i}=y\left(x_{i},\omega \right)+\varepsilon$$
where $t_{i}$ represents the observed value, and ε is noise. ε and ω, respectively, are assumed to be subject to Gaussian distribution $N\left(0,\sigma_{1}^{2}\right)$ and $N\left(0,\sigma_{2}^{2}\right)$. t is subject to Gaussian distribution with a mean of $y\left(x,\omega \right)$. The class conditional probability density function for t is given below.
(19)$$p\left(t|\omega \right)=\frac{1}{2\pi \sigma_{1}^{2}}exp\left(-\frac{1}{2\sigma_{1}^{2}}\overset{m}{\underset{i=1}{\sum }}{\left\{y\left(x_{i},\omega \right)-t_{i}\right\}}^{2}\right)$$

The prior probability of ω is given below.
(20)$$p\left(\omega \right)=\frac{1}{2\pi \sigma_{2}^{2}}exp\left(-\frac{1}{2\sigma_{2}^{2}}\omega^{T}\omega \right)$$

The following conclusion can be obtained according to the Bayesian rule.
(21)$$p\left(\omega |t\right)=\frac{p\left(\omega \right)p\left(t|\omega \right)}{p\left(t\right)}$$
(22)$$\ln\left(p\left(\omega |t\right)\right)=-\frac{1}{2\sigma_{1}^{2}}\overset{m}{\underset{i=1}{\sum }}{\left\{y\left(x_{i},\omega \right)-t_{i}\right\}}^{2}-\frac{1}{2\sigma_{2}^{2}}\omega^{T}\omega +c$$
where $p\left(\omega |t\right)$ is the posterior probability. $p\left(t\right)$ is a constant that is independent of ω. c is a constant. The prior probabilities correspond to the L2 regular term in the ridge regression. Hence, this algorithm is called BRR. BRR automatically introduces the regular term in the estimation process, and the result is the posterior distribution of the parameters. BRR avoids overfitting in maximum likelihood estimation and is able to obtain more precise parameter estimates.

### 5.4. Evaluation Model Establishment of MSL

The three cerebral blood oxygen characteristic parameters that passed the significance test are normalized. BRR is performed using the three normalized cerebral blood oxygen characteristic parameters as independent variables and the FMS scores of the subjects as dependent variables. Finally, the evaluation model of MSL is obtained as follows.
(23)$$MSL=1.39156\cdot C_{HbO}+1.528\cdot C_{\Delta Oxy}+0.94238\cdot \Delta COE-0.49018$$

Figure 5 shows the four-in-one plot of the residual analysis. It can be observed from the residual–fitted value scatter plot (c) that the scatter distribution in the figure has no obvious trend of trumpet shape or curve, indicating that the functional model fits the data well. The probability–residual plot (d) and the residual histogram (b) show that the residuals follow a normal distribution. The residual–observation plot (a) indicates that the overall data present stability without anomalies.

According to the MSSQ score, the eight subjects were divided into the non-sensitive group, the general group, and the sensitive group. The evaluation model was used to calculate the average motion sickness performance of each subject within 10 min. Through the Graybiel scale, the assessment scores of the subjects for their own physical symptoms were obtained, so as to reflect the degree of motion sickness of the subjects. Figure 6 reports the predicted curves of the average motion sickness degree and the evaluation scores of the Graybiel scale for the three groups of subjects with different sensitivities to motion sickness in the whole riding experiment. Figure 6 reports that the MSL predicted by the motion sickness evaluation model is positively correlated with the score of the Graybiel scale, and the higher the sensitivity of motion sickness is, the stronger the degree of motion sickness is. Therefore, the reliability of the motion sickness prediction model based on BRR is preliminarily verified.

## 6. Validation of Motion Sickness Evaluation Model from Vehicle Test

This study was supported by SAIC Motor R&D Innovation Headquarters. One driver, one experimenter, and two passengers were arranged by the Vehicle Integration Department to participate in the test to verify the effectiveness of the motion sickness evaluation model. The driver was arranged internally by SAIC Motor and they had a driving permit for the company’s internal vehicles. Two crew members were selected after being assessed by the MSSQ questionnaire. Passenger 1 belonged to the insensitive group of motion sickness. Passenger 2 belonged to the sensitive group of motion sickness.

Each subject participated in two rounds of the test. The duration of each round was 30 min. The mode of vehicle setup was different in the two rounds of testing. There were two modes of the experimental vehicle. One was normal mode, the other was comfortable mode. The comfort mode was optimized in the dynamic response and other links, which effectively reduced the sensory conflict of the participants.

The same driver was selected for all tests to ensure that the driver’s driving style for the vehicle was the same in each test. The experimental site was arranged in Anting Town, Jiading District, Shanghai. The driver chose random routes to drive (Figure 7). The experiment was scheduled in the afternoon. The cerebral blood oxygen signals of the two subjects were collected. Figure 7 shows the subjects and the monitoring interface of the experimenter in the vehicle test. In the experiment, the behavior of the subjects was not constrained. The subjects were free to browse content from their mobile phones and communicate during the experiment.

Table 2 reports the predicted values of MSL for two subjects in the normal mode and the comfortable mode. It can be seen from Table 2 that the predicted values of motion sickness in the comfortable mode were −0.490 and 3.272, respectively. The MSL prediction values of the two subjects in the normal mode were 2.170 and 3.279, respectively. That is to say, subjects achieved a low score of MSL in the comfort mode of the vehicle test. This is in line with expectations, and also validates the effectiveness of the motion sickness prediction model based on cerebral blood oxygen parameters.

## 7. Discussion

In this study, a driving simulation platform is built based on a six-degree-of-freedom parallel mechanism to simulate the road riding conditions. fNIRS technology is utilized to monitor the cerebral blood oxygen signals of the subjects in the riding simulation experiment. Three characteristic parameters with a high correlation with FMS values are extracted from cerebral blood oxygen signals. The prediction model of MSL is constructed by using the BRR algorithm, which eliminates the multicollinearity among independent variables. It can be observed from the coefficients of the evaluation model that ΔCOE is valid for characterizing the degree of motion sickness. This conclusion is consistent with the findings of Zhang et al. and Kayoko Yoshino et al. [26,37]. In this paper, a vehicle road test is arranged to validate the motion sickness evaluation model. The results show that the predicted values of MSL in the normal mode are significantly greater than the predicted values of motion sickness in the comfortable mode. This conclusion verifies the validity of the motion sickness evaluation model proposed in this study.

To explore the relationship between cerebral blood oxygen signals and gender difference in motion sickness, analysis of variance (ANOVA) is performed with gender as the independent variable and $C_{HbO}$, $C_{\Delta Oxy}$, and ΔCOE as the dependent variables. Table 3 shows that the difference between $C_{HbO}$ and gender is not significant (*p* = 0.26 > 0.05). The difference between ΔCOE and gender is also not significant (*p* = 0.14 > 0.05). However, the difference between $C_{\Delta Oxy}$ and gender is significant (*p* = 1.70 × 10^−5^ < 0.05). Therefore, $C_{\Delta Oxy}$ can be utilized as the main parameter for studying gender differences in motion sickness.

The importance of $C_{\Delta Oxy}$ can also be detected from the evaluation model. In the evaluation model, the coefficient of $C_{\Delta Oxy}$ is 1.528, which is higher than that of ΔCOE. This means that $C_{\Delta Oxy}$ has a stronger ability to characterize the degree of motion sickness than ΔCOE. $C_{\Delta Oxy}$ represents the difference between $C_{HbO}$ and $C_{HbR}$. However, the difference between $C_{HbO}$ and $C_{HbR}$ is related to the difference between $\Delta C_{HbO}$ and $\Delta C_{HbR}$, and $\Delta C_{ref}$ where $\Delta C_{ref}$ represents the difference between $C_{HbO}$ and $C_{HbR}$ in the reference state, as follows.
(24)$$C_{\Delta Oxy}=C_{HbO}-C_{HbR}=\Delta C_{HbO}-\Delta C_{HbR}+\Delta C_{ref}$$
where $\Delta C_{ref}$ has individual differences. However, ΔCOE is also related to the difference between $\Delta C_{HbO}$ and $\Delta C_{HbR}$. Thus, the above Equation (24) can be rewritten as follows.
(25)$$C_{\Delta Oxy}=-\sqrt{2}\Delta COE+\Delta C_{ref}$$

It can be concluded that the characteristic parameter $C_{\Delta Oxy}$ of fusion $\Delta C_{ref}$ has better motion sickness prediction ability than ΔCOE. This is of guiding significance for the construction of motion sickness models in the future.

Severe motion sickness may alter standing balance, reduce lower back function, and lead to changes in the expression of genes that play a role in osteogenesis, myogenesis, brain lymphatic development, inflammation, neuropathic pain, etc. [38]. The evaluation model of MSL proposed in this study can help prevent the harm caused by motion sickness. However, motion sickness cannot be completely avoided while riding. Many scholars have achieved the alleviation of motion sickness through the intervention of the environment or physiological intervention in the car. Hwang S et al. created an interface that adjusted its orientation in real time to match the orientation of the vehicle to help reduce motion sickness among passengers while looking at the screen of a digital device [39]. Kim H et al. determined the effect of four fragrance factors on the prevention of motion sickness through EEG and MISC data [40].

People generally assume that passengers are more susceptible to motion sickness than drivers. However, there have not been many research studies that compare the neural activity of passengers and drivers together with motion sickness. Li et al. concluded from their study that the enhanced alpha instant signal activation of passengers was due to a higher degree of motion sickness. Therefore, compared with drivers, passengers experience more conflicts of multimodal sensory systems, which require neurophysiological regulation [41]. However, that does not mean drivers do not experience motion sickness. If the driver has motion sickness, it can easily lead to traffic accidents, which are irreversible. Therefore, the study of drivers’ motion sickness is also a direction worthy of further study.

There are many aspects to be improved in any future study. First, riding simulation should not only simulate the physical feeling of riding brought about by the actual road conditions but also simulate the actual riding state from the visual and auditory aspects. Visually induced motion sickness and audially induced motion sickness are also factors that cannot be ignored in actual riding. Therefore, in the follow-up work, it is necessary to build a complete driving simulation environment. Second, in the process of subject recruitment, subjects in more age groups are required. In this study, the subjects were all students. The performance of these subjects can only represent the performance of motion sickness in a part of the age group. Therefore, in the follow-up work, it is necessary to recruit more subjects of different age groups to ensure the universality of the obtained evaluation model of MSL.

## 8. Conclusions

The purpose of this study is to investigate the correlation model between the degree of motion sickness and cerebral blood oxygen signals during riding. Therefore, based on the 6-DOF parallel mechanism, a driving simulation platform is built to simulate the actual riding conditions on the road. The continuous acquisition of cerebral blood oxygen signals can reflect the physiological state of subjects in real time. In this study, fNIRS technology is applied to monitor the cerebral blood oxygen signals of the subjects in the riding simulation experiment. The cerebral blood oxygen signals are then filtered and motion artifacts are suppressed. Three parameters with a high correlation with FMS values are extracted from the cerebral blood oxygen signals. In this study, the BRR algorithm is used to construct an evaluation model for the degree of motion sickness, which eliminates the multicollinearity among the independent variables and improves the reliability of the prediction model. The Graybiel scale score is utilized to preliminarily verify the prediction effect of the model. Random road conditions and two driving modes are selected to carry out the verification vehicle test. These two driving modes are ordinary mode and comfortable mode, respectively, which can cause different degrees of motion sickness effects in the subjects. The experimental results show that the predicted values of motion sickness in the normal mode are 2.169 and 3.279, respectively, which are significantly larger than the predicted values of motion sickness in the comfortable mode of −0.490 and 3.272. This conclusion is in line with expectations. Therefore, the motion sickness prediction model proposed in this study is effectively verified. This provides a reference for the evaluation and prediction of motion sickness.

## Figures and Tables

**Figure 1 diagnostics-13-01403-f001:**
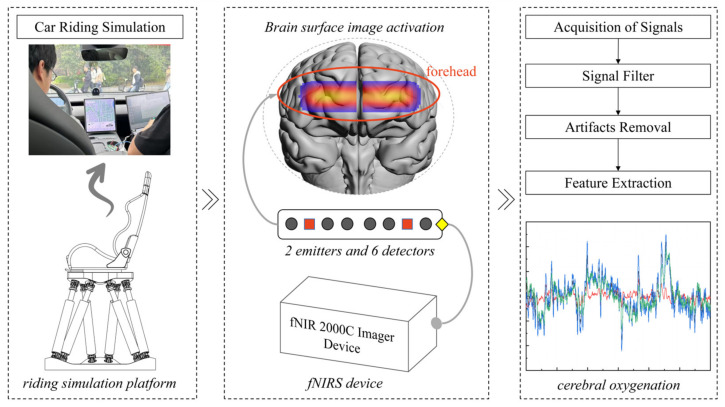
Experimental flow design based on riding simulator.

**Figure 2 diagnostics-13-01403-f002:**
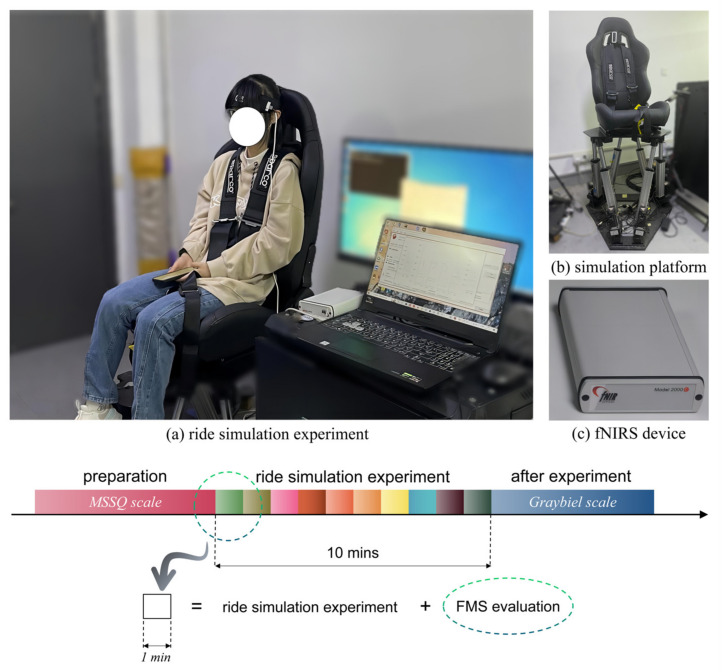
Experimental equipment and testing process. (**a**) ride simulation experiment; (**b**) simulation platform; (**c**) fNIRS device.

**Figure 3 diagnostics-13-01403-f003:**
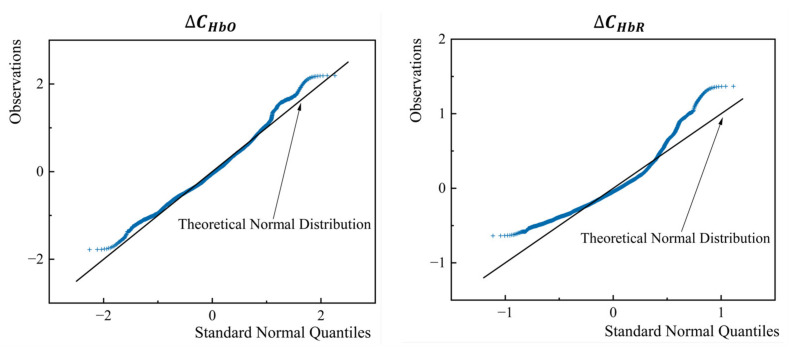
Normality test of signals (one subject).

**Figure 4 diagnostics-13-01403-f004:**
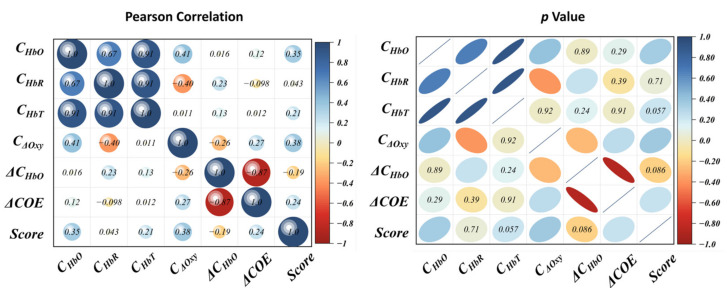
Pearson correlation analysis and significance test for independent and dependent variables.

**Figure 5 diagnostics-13-01403-f005:**
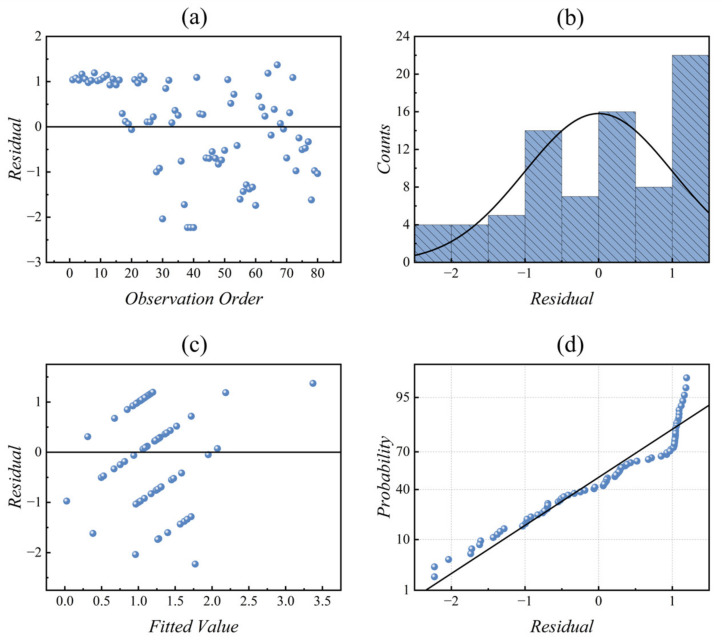
Four-in-one plot for residual analysis. (**a**) Residual–observation plot; (**b**) Residual histogram; (**c**) Residual–fitted value scatter plot; (**d**) Probability–residual plot.

**Figure 6 diagnostics-13-01403-f006:**
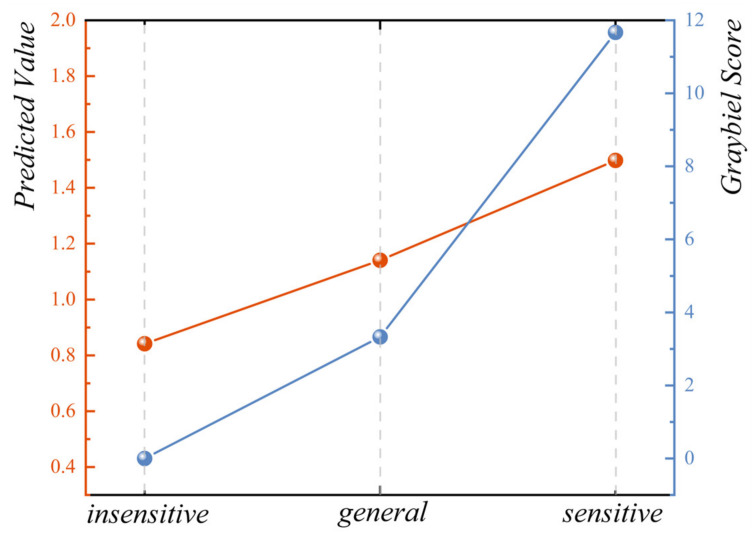
Cross-validation of predicted motion sickness value and Graybiel score.

**Figure 7 diagnostics-13-01403-f007:**
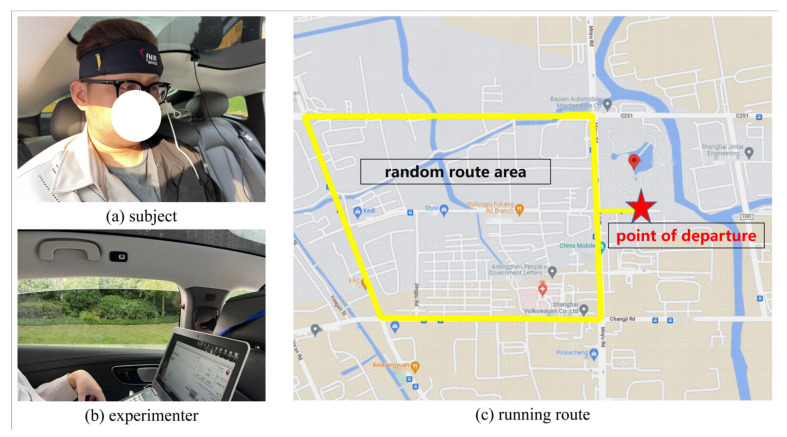
Vehicle test and random route area. (**a**) subject; (**b**) experimenter; (**c**) running route.

**Table 1 diagnostics-13-01403-t001:** Parameters of the 6-degree-of-freedom platform.

Parameters	Value
CylinderOrigLength	580.000000
MaxTravelRange	295.000000
TopHexagonLongerSideLength	420.000000
TopHexagonShortSideLength	120.000000
TopCircumcircleDiamiter	567.000000
BottomHexagonLongerSideLength	450.000000
BottomHexagonShortSideLength	150.000000
BottomCircumcircleDiamiter	634.000000
PlatformMaxYTravelRange	304.189148
PlatformMaxRotateAngle	32.444931

**Table 2 diagnostics-13-01403-t002:** The predicted MSL value of the subjects during the vehicle test.

Subject Number	Normal Mode	Comfortable Mode
1	2.170	−0.490
2	3.279	3.272

**Table 3 diagnostics-13-01403-t003:** ANOVA on gender and three cerebral oxygen characteristics.

	DF	SS	MS	F	*p*
$C_{HbO}$					
Model	1	4.13	4.13	1.28	0.26
Error	78	250.94	3.22		
Total	79	255.08			
$C_{\Delta Oxy}$					
Model	1	35.29	35.29	21.02	1.70 × 10^−5^
Error	78	130.91	1.68		
Total	79	166.19			
ΔCOE					
Model	1	4.05 × 10^−5^	4.05 × 10^−5^	2.19	0.14
Error	78	1.44 × 10^−3^	1.84 × 10^−5^		
Total	79	1.48 × 10^−3^			

Note: DF—degree of freedom; SS—sum of squares of deviation from mean; MS—mean square.

## Data Availability

All data included in this study are available upon request by contact with the corresponding author. The data are not publicly available because of ethical restrictions.

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
