# Peer review of "Assessing Passengers’ Motion Sickness Levels Based on Cerebral Blood Oxygen Signals and Simulation of Actual Ride Sensation"

_diagnostics, 2023, doi:10.3390/diagnostics13081403_

Round 1
Reviewer 1 Report
This paper studies the motion sickness evaluation model for passengers based on multiple characteristic parameters of cerebral blood oxygen signals. The riding simulation platform and the functional near-infrared spectroscopy technology are utilized to monitor the cerebral blood oxygen signals. Bayesian ridge regression algorithm is applied to construct an assessment model. The paper is well-written. There is a suggestion that the parameters of the 6 Dof simulation platform can be given.
Author Response
Thanks to the Reviewer for valuable suggestions. All changes have been highlighted in the paper.
Comment 1: There is a suggestion that the parameters of the 6 Dof simulation platform can be given.
Response: Thanks for the suggestion. The parameters of the 6 DOF simulation platform are listed in Table 1. The changes to address this issue are listed below.
Table 1. Parameters of the 6-degree-of-freedom platform.
Parameters |
Value |
CylinderOrigLength |
580.000000 |
MaxTravelRange |
295.000000 |
TopHexagonLongerSideLength |
420.000000 |
TopHexagonShortSideLength |
120.000000 |
TopCircumcircleDiamiter |
567.000000 |
BottomHexagonLongerSideLength |
450.000000 |
BottomHexagonShortSideLength |
150.000000 |
BottomCircumcircleDiamiter |
634.000000 |
PlatformMaxYTravelRange |
304.189148 |
PlatformMaxRotateAngle |
32.444931 |

Reviewer 2 Report
This manuscript addresses an interesting subject. However, some issues are raised that have to be solved:
1) The population of the experiment is very low: only four subjects. Can you discuss if you can really obtain reliable conclusions from this?.
2) The population is two males and two females. Did you detect any difference because of the gender? (in any case, it is difficult from this low population).
3) In Figure 6, the difference between the predicted value and the Graybiel score is high in the case of the general group (close to 50%). Discuss which differences are acceptable in this curve and why.
4) Typos:
"Pearson" in Figures 3 and 4.
Upper case in the first letter of the first word of all the Subsections: 2.1, 2.2, 2.3, 2.4, 5.1.
Author Response
Thanks to the Reviewer for valuable suggestions. All changes have been highlighted in the paper.
Comment 1: The population of the experiment is very low: only four subjects. Can you discuss if you can really obtain reliable conclusions from this?
Response: Thanks for the suggestion. The experimental requirement is to recruit subjects with three different kinds of sensitivities to motion sickness (insensitive, general and sensitive). The four subjects recruited at first fit this requirement, so we started to analyze the data. But as you suggested, four subjects are far from enough to rule out individual differences. Therefore, we recruit four additional subjects to supplement the data. The cerebral blood oxygen signals of a total of 8 subjects can improve the convincing of the motion sickness evaluation model. Relevant images, data and formulas have been updated in the paper. The changes to address this issue are listed below.
Eight subjects are recruited in this study. These eight subjects are undergraduate students from Shanghai University, including four male students and four female students. Informed consent is obtained from the eight subjects before conducting the experiment.
Comment 2: The population is two males and two females. Did you detect any difference because of the gender? (in any case, it is difficult from this low population).
Response: Thanks for the suggestion. Gender differences do lead to differences in motion sickness recognition and evaluation methods. It is important to obtain the cerebral blood oxygen parameters which are closely associated with gender differences. Therefore, following your suggestion, we explore the significance between the three variables from the evaluation model and gender. Finally, it is found that the difference between and gender is significant. Therefore, can be used to carry out the study of gender differences in motion sickness. The changes to address this issue are listed below.
To explore the relationship between cerebral blood oxygen signals and gender difference in motion sickness, analysis of variance (ANOVA) is performed with gender as the independent variable and , , as the dependent variable. Table 3 shows that the difference between and gender is not significant (P=0.26>0.05). The difference between and gender is also not significant (P=0.14>0.05). However, the difference between and gender is significant (P=1.70E-05<0.05). Therefore, can be utilized as the main parameter for studying gender differences in motion sickness.
Table 3. ANOVA on gender and three cerebral oxygen characteristics.
|
DF |
SS |
MS |
F |
P |
Model |
1 |
4.13 |
4.13 |
1.28 |
0.26 |
Error |
78 |
250.94 |
3.22 |
|
|
Total |
79 |
255.08 |
|
|
|
Model |
1 |
35.29 |
35.29 |
21.02 |
1.70E-05 |
Error |
78 |
130.91 |
1.68 |
|
|
Total |
79 |
166.19 |
|
|
|
Model |
1 |
4.05E-05 |
4.05E-05 |
2.19 |
0.14 |
Error |
78 |
1.44E-03 |
1.84E-05 |
|
|
Total |
79 |
1.48E-03 |
|
|
|
Note: DF-degree of freedom; SS-sum of squares of deviation from mean; MS-mean square
Comment 3: In Figure 6, the difference between the predicted value and the Graybiel score is high in the case of the general group (close to 50%). Discuss which differences are acceptable in this curve and why.
Response: Thanks for the suggestion. Here, the subjects were divided into three groups: sensitive, general, and insensitive to motion sickness. Graybiel score was treated as a reference to preliminarily verify the effectiveness of the evaluation model in this paper. Graybiel score can distinguish the degree of motion sickness in three different groups of people. And with the increase of motion sickness sensitivity, Graybiel score also increases. As long as the predicted values of the evaluation model also show the same trend as the Graybiel score, the ability of the evaluation model to identify the degree of motion sickness can then be preliminarily verified. As shown in Figure 6, the predicted value of motion sickness degree and Graybiel score keep increasing together, so the effectiveness of the evaluation model is preliminarily verified. The changes to address this issue are listed below.
Figure 6. Cross-validation of predicted motion sickness value and Graybiel score.
Comment 4: Typos: "Pearson" in Figures 3 and 4. Upper case in the first letter of the first word of all the Subsections: 2.1, 2.2, 2.3, 2.4, 5.1.
Response: Thanks for the suggestion. We are very sorry to make a mistake in Figure 3. Now Figure 3 is corrected. The changes to address this issue are listed below.
Figure 3. Normality test of signals (one subject).
Figure 4. Pearson correlation analysis and significance test for independent and dependent variables.
2.1. Definition and cause of motion sickness
2.2. Recognition and classification of motion sickness
2.3. Motion sickness recognition based on cerebral blood oxygen signals
2.4. Motivation and structure of this study
5.1. Extraction of multiple characteristic parameters from cerebral blood oxygen signals

Reviewer 3 Report
This well-written manuscript paper is quite acceptable “as-is” and offers some valuable insights to the field of motion sickness and diagnostic methods. The methods are well desscribed and results clearly presented.
Several points that the authors may consider.
[Lines 130-135] It is at this point that the authors introduce the significance of the work in that the “…this study restores the actual ride sensation through the ride simulation platform. The relationship between motion sickness and cerebral blood oxygen signals is studied by simulating real riding conditions.” Unfortunately the significance of this is lost in the introduction and, further you would not know from the title the essence of this. Some change in the title may be justified, as in the current for the title is no entirely engaging and understates the value of reading the manuscript
[Lines 136-146] Excellent introduction and REALLY helped with wayfinding through the manuscript.
[Lines165-167] It is recognized that the four (4) subjects created 40 data sets, but it would help to discuss in more detail the reasoning for such a small N and thus this might just be a pilot study and not set to have definitive statistical outcomes.
[Lines263-275] Possibly an introductory table of the parameters and whether they were data-driven or calculation outcomes (i.e., eqns 12-14). Yes, you do provide the equations and it is obvious, but at first reading, line 266 makes the reader pause. Is there apriority? A specific hypothesis? The study subsequently identifies on Lines 281-282 what I believe is the significant outcome. Not sure how all of this could be edited for clarity and focus to help the reader understand what is most important.
[Lines303-304] Am not sure if “…it can be seen from Figure 3 … normal distribution…” as this seems to be more of an inferred outcome. Add just a bit of descriptive test to persuade the reader?
[Lines 320-321] Excellent ending to the section, but the reader is left “hanging” on the final sentence that “So it is necessary to avoid multicollinearity in the regression process.” It might be helpful to include a line to cue the reader that the multicollinearity issue will be addressed in the section that immediately follows.
[Line 365] Replace the copyright symbol in the text. And, do the four linear lines correspond with the four subjects? What is the implication of Figure c?
[Line 449-450] This is an excellent outcome but, again, the title focuses on the six characteristics that does not draw in the reader.
Author Response
Thanks to the Reviewer for valuable suggestions. All changes have been highlighted in the paper.
Comment 1: [Lines 130-135] It is at this point that the authors introduce the significance of the work in that the “…this study restores the actual ride sensation through the ride simulation platform. The relationship between motion sickness and cerebral blood oxygen signals is studied by simulating real riding conditions.” Unfortunately the significance of this is lost in the introduction and, further you would not know from the title the essence of this. Some change in the title may be justified, as in the current for the title is no entirely engaging and understates the value of reading the manuscript.
Response: Thanks for the suggestion. As you mentioned, the title is not enough to illustrate the essence of the simulation platform that it provides the simulation of actual ride sensation. So we revise the title. We also put some description of the simulation platform in the introduction. The changes to address this issue are listed below.
Assessing passengers' motion sickness levels based on cerebral blood oxygen signals and simulation of actual ride sensation
In this study, a ride simulation platform is built to restore the actual ride sensation. The measurement method of cerebral blood oxygen signals from motion sickness is established by near-infrared equipment to quantitatively evaluate and predict MSL.
Comment 2: [Lines 136-146] Excellent introduction and REALLY helped with wayfinding through the manuscript.
Response: Thanks for the commendation. We summarize the content of each chapter and form the general structure of the article, which helps readers to understand.
Comment 3: [Lines165-167] It is recognized that the four (4) subjects created 40 data sets, but it would help to discuss in more detail the reasoning for such a small N and thus this might just be a pilot study and not set to have definitive statistical outcomes.
Response: Thanks for the suggestion. As you suggested, four subjects are far from enough to rule out individual differences. This might be a pilot study and not set to have definitive statistical outcomes. Therefore, we recruit four additional subjects to supplement the data. The cerebral blood oxygen signals of a total of 8 subjects creating 80 data sets can improve the convincing of the motion sickness evaluation model. Relevant images, data and formulas have been updated in the paper. The changes to address this issue are listed below.
Eight subjects are recruited in this study. These eight subjects are undergraduate students from Shanghai University, including four male students and four female students. Informed consent is obtained from the eight subjects before conducting the experiment.
Comment 4: [Lines263-275] Possibly an introductory table of the parameters and whether they were data-driven or calculation outcomes (i.e., eqns 12-14). Yes, you do provide the equations and it is obvious, but at first reading, line 266 makes the reader pause. Is there apriority? A specific hypothesis? The study subsequently identifies on Lines 281-282 what I believe is the significant outcome. Not sure how all of this could be edited for clarity and focus to help the reader understand what is most important.
Response: Thanks for the suggestion. As you suggested, the original description of the six parameters is not clear, which is easy to cause readers' incomprehension. Therefore, we modified this to make the introduction of these six parameters become logical. We first introduce the most basic parameter . Then we introduce and , which are based on . Then and are introduced, and they are calculated on the basis of and . Finally, the parameter is introduced through previous studies. The changes to address this issue are listed below.
is obtained directly from formula (10), which is calculated from the light intensity signals by Beer-Lambert law.
and represent the sum and difference of and respectively.
Comment 5: [Lines303-304] Am not sure if “…it can be seen from Figure 3 … normal distribution…” as this seems to be more of an inferred outcome. Add just a bit of descriptive test to persuade the reader?
Response: Thanks for the suggestion. We are very sorry to make a mistake in Figure 3. Now Figure 3 is corrected. And an explanation is supplemented in the paper. The changes to address this issue are listed below.
It can be seen from Figure 3 that the cerebral blood oxygen signals are subject to the normal distribution (the curves of the parameters are close to the theoretical normal distribution curves).
Figure 3. Normality test of signals (one subject).
Comment 6: [Lines 320-321] Excellent ending to the section, but the reader is left “hanging” on the final sentence that “So it is necessary to avoid multicollinearity in the regression process.” It might be helpful to include a line to cue the reader that the multicollinearity issue will be addressed in the section that immediately follows.
Response: Thanks for the suggestion. We have include a line in 5.2. The changes to address this issue are listed below.
This paper introduces a Bayesian ridge regression algorithm to address the multicollinearity issue.
Comment 7: [Line 365] Replace the copyright symbol in the text. And, do the four linear lines correspond with the four subjects? What is the implication of Figure c?
Response: Thanks for the suggestion. We have replaced the copyright symbol in the text. The four-in one plot is a way of residual analysis, which is independent of the numbers of the subjects. Figure (c) is utilized to test whether the variances of the residuals are the same for all fitted values. The scatter distribution in figure (c) has no obvious trend of trumpet shape and curve, indicating that the functional model fits the data well. The changes to address this issue are listed below.
It can be observed from the residual-fitted value scatter plot (c) that the scatter distribution in the figure has no obvious trend of trumpet shape or curve, indicating that the functional model fits the data well.
Comment 8: [Line 449-450] This is an excellent outcome but, again, the title focuses on the six characteristics that does not draw in the reader.
Response: Thanks for the suggestion. We have revised the title. The changes to address this issue are listed below.
Assessing passengers' motion sickness levels based on cerebral blood oxygen signals and simulation of actual ride sensation

Round 2
Reviewer 2 Report
The issues that I raised have been answered. This manuscript can be published.